# Current Controversies on Adequate Circulating Vitamin D Levels in CKD

**DOI:** 10.3390/ijms27010108

**Published:** 2025-12-22

**Authors:** Adriana S. Dusso, Daniela J. Porta, Carlos Bernal-Mizrachi

**Affiliations:** 1Division of Endocrinology, Metabolism and Lipid Research, Washington University School of Medicine, St. Louis, MO 63110, USA; 2Department of Medicine, VA Medical Center, St. Louis, MO 63106, USA

**Keywords:** vitamin D, calcidiol, calcitriol, FGF23, klotho, renal damage, cardiovascular damage

## Abstract

Management of secondary hyperparathyroidism (SHPT) in chronic kidney disease (CKD) has evolved dramatically over the past five decades, driven by discoveries that have fundamentally reshaped our understanding of the vitamin D endocrine system and its role in disease progression. This review synthesizes the key pathophysiological insights and clinical evidence underlying three critical paradigm shifts. The first shift moved beyond simple calcitriol replacement with the development of selective vitamin D receptor activators (VDRAs) designed to minimize hypercalcemia while maximizing PTH suppression. Crucially, these analogs revealed unexpected survival benefits, suggesting protective VDR actions extending beyond mineral metabolism. The second shift recognized the profound prevalence and independent mortality risk associated with nutritional vitamin D (25(OH)D) deficiency in CKD. This highlighted the kidney’s complex role in maintaining systemic 25(OH)D supply and the importance of extrarenal vitamin D activation, although optimal assessment, targets, and supplementation strategies remain highly controversial due to CKD-specific pathophysiology (e.g., megalin loss, impaired uptake, obesity effects) and complex dosing paradoxes. The third, and most impactful, shift centers on the FGF23-Klotho axis. Pathologically high FGF23 is now established as a direct cardiovascular and skeletal toxin, acting via Klotho-independent pathways in CKD, while the profound deficiency of the protective, anti-aging hormone Klotho exacerbates systemic damage (inflammation, oxidative stress, impaired autophagy). This creates a major therapeutic dilemma, as VDRAs induce protective Klotho but worsen toxic FGF23, while calcimimetics do not increase FGF23 but offer no Klotho benefit. Furthermore, this complex interplay is obscured by significant limitations in accurately measuring FGF23 isoforms, soluble Klotho, and true vitamin D status. These paradigm shifts reveal a complex pathophysiology far beyond simple PTH control, demanding a move towards nuanced, potentially combined therapeutic strategies that balance FGF23 burden with Klotho preservation. Overcoming the profound diagnostic limitations to accurately monitor this axis and guide personalized therapy represents the critical next frontier in improving outcomes for patients with CKD.

## 1. Introduction

Chronic Kidney Disease (CKD), affecting over 11% of the global population, is characterized by accelerated multi-organ aging that escalates the risk of cardiovascular morbidity and mortality by 10- to 20-fold compared to the general population [1,2].

The secondary hyperparathyroidism (SHPT) in chronic kidney disease (CKD), characterized by elevated parathyroid hormone (PTH), parathyroid gland overgrowth, and disordered mineral metabolism, is a major contributor to the high rates of bone disease and cardiovascular mortality in CKD patients [3].

For over fifty years, since the discovery of the kidney’s role in synthesizing active vitamin D (1,25(OH)_2_D) [4], the management of secondary hyperparathyroidism (SHPT) in chronic kidney disease (CKD) has been a central clinical challenge. The initial strategy of simple hormone replacement [5] has evolved dramatically, shaped by decades of research and clinical experience. This evolution is best understood through three critical paradigm shifts that have redefined our approach to using nutritional and active vitamin D interventions in CKD. This review synthesizes the major paradigm shifts in vitamin D biology, focusing on how (1) new pathophysiological discoveries have reshaped our understanding of CKD progression and (2) the compelling case these findings make for re-evaluating current therapeutic recommendations.

## 2. Paradigm Shift 1: From Simple 1,25(OH)_2_D Replacement to Selective VDR Activation

The initial use of the hormonal form of vitamin D, 1,25(OH)_2_D or calcitriol, to suppress PTH was hampered by the frequent side effects of the hypercalcemia and hyperphosphatemia resulting from the development of parathyroid resistance to calcitriol [6,7]. This led to the first major shift: the development of synthetic vitamin D analogs (e.g., paricalcitol). These molecules were designed to be “selective,” potently suppressing the PTH gene while having less impact on intestinal calcium and phosphate absorption. Importantly, large observational studies revealed that patients treated with these analogs had a significant survival advantage over those treated with calcitriol [8]. This finding was a crucial revelation, suggesting that the benefits of activating the Vitamin D Receptor (VDR) extended far beyond simple PTH control to include direct, “pleiotropic” protective effects on the cardiovascular system.

### 2.1. Pathophysiological Discoveries and Impact on CKD Progression

#### 2.1.1. Parathyroid Glands

Active vitamin D exerts control over the secondary hyperparathyroidism of CKD through several key mechanisms:(a)Direct Gene Suppression: The active vitamin D/Vitamin D Receptor (VDR) complex directly suppresses the transcription of the *PTH* gene, a foundational mechanism of control [9].(b)Sensitization to Calcium: Active vitamin D is crucial for maintaining the parathyroid’s sensitivity to calcium. It achieves this by upregulating the expression of the Calcium-Sensing Receptor (CaSR) [10,11,12]. In CKD, a lack of adequate active vitamin D leads to fewer CaSRs, making the gland resistant to PTH suppression by calcium. This is particularly important because the CaSR also functions as a phosphate sensor [13]; high phosphate levels in CKD inhibit the CaSR activity, further stimulating PTH secretion. By increasing CaSR expression, active vitamin D helps counteract this phosphate-driven stimulation.(c)Inhibition of Parathyroid Growth: Uncontrolled SHPT results from parathyroid hyperplasia. This overgrowth is driven by increases in cyclooxygenase2-prostaglandin E2 and mTOR [14], and also by a powerful autocrine growth loop involving Transforming Growth Factor-alpha (TGF-α) and its receptor, EGFR, which is potently exacerbated by the hyperphosphatemia of advanced CKD [7,15]. The active vitamin D/VDR complex directly counters this vicious cycle at its source by suppressing ADAM17 [16], the enzyme that initiates the loop through the release of TGF-α from the parathyroid cell surface.While vitamin D also influences other growth regulators like p21 and p27 [17,18,19], this dual action on the ADAM17 pathway is a central mechanism for overcoming the most severe nodular forms of SHPT.(d)Parathyroid Desensitization to VDR Actions: Critically, this enhanced ADAM17/TGF-α/EGFR axis does more than just stimulate growth; it is also the primary driver of vitamin D resistance in CKD. It achieves this by decreasing the cellular C/EBPβ/LIP ratio, which in turn leads to a marked suppression of the VDR gene itself, the cause of resistance to active vitamin D therapy in advanced CKD [7]. Therefore, by inhibiting the dominant, phosphate-driven ADAM17/TGF-α/EGFR loop, active vitamin D simultaneously controls parathyroid cell proliferation and preserves the gland’s essential sensitivity to active vitamin D. However, the pathological importance of the C/EBPβ mechanism extends beyond the parathyroid gland, driving systemic functional vitamin D deficiency. Evidence for this link comes from models demonstrating that inflammatory challenges (e.g., LPS stimulation) promote LIP synthesis [20], which then directly suppresses the VDR gene. Furthermore, LIP accumulation has been implicated in favoring ER stress-driven apoptosis [20]—a critical pathway for non-skeletal damage (e.g., vascular calcification propensity) in the CKD milieu [21].

The resulting VDR downregulation causes profound cellular resistance, which blunts the anti-inflammatory actions of vitamin D. This pathway is further complicated by the fact that active vitamin D itself induces C/EBPβ expression [22], suggesting a complex cross-talk that modulates the very resistance mechanism it attempts to overcome. This combined mechanism contributes significantly to the persistent inflammation that drives CKD progression.

#### 2.1.2. Bone

Beyond the powerful influence of active vitamin D on bone health through the effective control of secondary hyperparathyroidism, the active vitamin D/VDR complex exerts crucial direct, PTH-independent effects on bone cells to maintain skeletal integrity. Specifically, on bone remodeling: Active vitamin D is a master regulator of genes that control bone formation and resorption, including RANKL, osteoprotegerin (OPG), and osteocalcin (Reviewed in [23]). In CKD, the deficiency of active vitamin D, combined with resistance to its actions, disrupts this delicate balance of bone turnover.

#### 2.1.3. Systemic Protection

Beyond its role in mineral metabolism, the Vitamin D Receptor (VDR) acts as a master regulator of hundreds of genes critical for cellular survival. This extends to powerful protective actions that directly combat the systemic damage seen in CKD, including hypertension and inflammation.

(a)Control of the Renin–Angiotensin–Aldosterone System (RAAS)

A primary non-classical benefit of active vitamin D is its multifaceted control of the RAAS. The active vitamin D/VDR complex exerts this control through two distinct mechanisms:

a.1. Direct Renin Suppression: It directly suppresses the transcription of the *renin* gene, mitigating the initial step of RAAS activation [24].

a.2. Induction of the Counter-Regulatory Axis: VDR activation also increases the cell membrane expression of Angiotensin-Converting Enzyme 2 (ACE2) [25]. This key enzyme provides a protective counterbalance by converting Angiotensin II to Angiotensin-(1–7), which then activates the Mas receptor to exert potent anti-hypertensive-, anti-fibrotic-, and anti-inflammatory effects [26,27,28,29].

(b)Renal and Vascular Protection:

The main effector of RAAS, Angiotensin II, drives renal and vascular damage by activating the pro-fibrotic and pro-inflammatory ADAM17/TGF-α/EGFR loop, leading to glomerulosclerosis, proteinuria, and renal inflammatory cell infiltration [30]. In the vasculature, active vitamin D directly counteracts this at a molecular level. VDR activation induces microRNA-145 (miR-145) [31], a master regulator of the vascular smooth muscle cell (VSMC) phenotype [32,33], that is also known to suppress ADAM17 [34]. This mechanism is critical for vascular health; in uremia, the impaired induction of miR-145 causes VSMCs to differentiate into bone-like cells, a key driver of medial arterial calcification, or the “hardening” of arteries [35].

(c)Anti-Inflammatory Effects:

In addition to its major effects in innate and acquired immunity [36,37], the active vitamin D/VDR complex combats the chronic inflammation of CKD through a dual mechanism targeting TNFα. First, it directly suppresses the transcription of the *TNFα* gene, reducing the synthesis of this key pro-inflammatory cytokine at its source [38]. Second, by inducing miR-145 to suppress ADAM17, it also reduces the shedding and release of any existing TNFα from the cell surface [34]. This two-pronged approach is crucial for breaking a vicious inflammatory loop where TNFα itself can stimulate more ADAM17 expression [39], thus preventing the amplification of systemic, immune cell-driven damage.

These newly understood, systemic protective effects of VDR activation are summarized in Figure 1.

### 2.2. Therapeutic Implications and Recommendations

#### The Active Vitamin D Therapeutic Paradox: From Promising Mechanistic Endpoints to Failed Outcome Trials

Early research sparked considerable enthusiasm for the protective effects of active vitamin D. Powerful synergistic effects were demonstrated when a vitamin D analog was added to standard RAAS blockade [40,41,42]. By attacking the renin-angiotensin system from two different points, this dual therapy achieved a more potent reduction in proteinuria and blood pressure, raising hopes for a major breakthrough in cardiovascular and renal protection.

However, this initial promise did not translate to success in larger, more definitive trials designed to test or hard clinical endpoints. Despite Mendelian Randomization studies linking vitamin D deficiency with higher blood pressure and hypertension risk [43] and the demonstration from the UK Biobank that 25(OH)D levels below 10 ng/mL significantly increased the risk of all cause and cause specific mortality [44], subsequent studies, including PRIMO [45] and J-DAVID [46], ended in frustration. Although both trials confirmed successful biochemical control of SHPT—with PRIMO achieving significant PTH suppression and J-DAVID enrolling patients with already controlled PTH levels (80 pg/mL)—they consistently showed no benefit in reducing left ventricular hypertrophy, major adverse cardiovascular events, or overall mortality.

This created a frustrating disconnect between promising early results and failed major outcomes (Reviewed in [47]). The resolution to this paradox would be found in the third major paradigm shift, which revealed Fibroblast Growth Factor 23 (FGF23) as the dominant, and previously unrecognized, driver of cardiovascular risk in CKD.

Reflecting the consensus of this era, successive KDIGO guidelines established active vitamin D (calcitriol) and its analogs as the cornerstone, standard-of-care for the management of SHPT, as this was long before the clinical focus shifted to FGF23 or the introduction of calcimimetics.

## 3. Paradigm Shift 2: The Rise in Nutritional Vitamin D and Systemic Health

The second paradigm shift was the recognition that nutritional vitamin D deficiency (measured by circulating 25-hydroxyvitamin D below 20 ng/mL) [48], and not just calcitriol deficiency, is highly prevalent across all CKD stages and is an independent predictor of mortality and disease progression [49,50,51]. This realization highlighted that the kidney’s role extends beyond simply activating 25(OH)D to calcitriol; it is also central to maintaining the body’s overall vitamin D supply.

### 3.1. Pathophysiological Discoveries and Impact on CKD Progression

CKD drives nutritional vitamin D deficiency through several distinct mechanisms:(a)Impaired Initial Synthesis (Skin and Liver): CKD contributes to lower overall vitamin D levels by reducing the cutaneous synthesis of cholecalciferol (Vitamin D3) in the skin [52], which can be exacerbated by factors such as azotemia or limited sun exposure. Furthermore, there is evidence that CKD leads to abnormal hepatic conversion of cholecalciferol to 25(OH)D (25-hydroxylation), impairing the essential first step of metabolic activation.(b)Loss of Renal 25(OH)D Recycling (Megalin Failure): A critical early event in CKD is the loss of the endocytosis receptor megalin in the proximal tubules [53]. Megalin is essential for reabsorbing the filtered 25(OH)D bound to its vitamin D binding protein (DBP) [54]. Its failure has a dual consequence: it reduces the substrate available for renal calcitriol synthesis and, just as importantly, prevents the recycling of 25(OH)D back into circulation. This systemic loss of substrate starves the rest of the body’s tissues of the 25(OH)D they need, crippling their ability to perform local calcitriol synthesis and compromising the protective effects of autocrine/paracrine VDR activation.(c)Impaired Extrarenal 25(OH)D Uptake: The problem of 25(OH)D deficiency in CKD extends beyond defective renal recycling; extrarenal tissues also show a marked defect in their ability to take up the prohormone. Immune cells from dialysis patients, for example, demonstrate this impaired uptake [55]. This suggests that even if circulating 25(OH)D levels are adequate, the tissues that depend on it for local calcitriol synthesis cannot access it, thereby crippling the protective autocrine/paracrine benefits of VDR activation essential for slowing CKD progression. The critical importance of these extrarenal pathways is underscored by findings in anephric individuals: they retain the ability to produce significant amounts of calcitriol, and can even normalize circulating levels, provided they are given sufficient 25(OH)D [56].(d)Therapy-Induced Catabolism: Paradoxically, the high doses of active vitamin D/analogs used to treat SHPT can worsen nutritional vitamin D deficiency. These drugs potently induce CYP24A1 [57], the enzyme that degrades both active vitamin D and its precursor, 25(OH)D. This creates a vicious cycle where the treatment for one aspect of the disease exacerbates the underlying systemic deficiency.

This highlighted the kidney’s crucial role in maintaining systemic vitamin D status and underscored the importance of extra-renal vitamin D activation of 25(OH)D to calcitriol for VDR autocrine/paracrine pro-survival functions in tissues throughout the body. Critically, despite their higher potency for VDR activation, neither calcitriol nor its analogs can correct nutritional vitamin D deficiency.

### 3.2. Therapeutic Implications and Recommendations

A central challenge in clinical practice stems from the 2017 KDIGO guideline, which recommends correcting nutritional vitamin D deficiency in CKD patients “as for the general population [58].” This deferral is a major limitation, as guidelines for the general population—such as the recent 2024 Endocrine Society update [59]—are not designed to address the unique and complex pathophysiology of CKD. These general recommendations fail to account for the specific pathogenic mechanisms of CKD (e.g., impaired megalin-mediated 25(OH)D recycling, defective extrarenal uptake) that render a simple “general population” approach insufficient. Furthermore, this strategy relies on standard serum 25(OH)D thresholds (e.g., <20 ng/mL for deficiency, 20–29 ng/mL for insufficiency, and >30 ng/mL for sufficiency) that are themselves subjects of intense debate. These static cut-offs are of questionable validity even in healthy individuals and are particularly ill-suited for the complex metabolic disturbances of a CKD patient. While a 2018 NKF consensus conference [60] provided a more nuanced, tiered approach for CKD patients (e.g., intervening at levels < 20 ng/mL), a clear, evidence-based target for supplementation in this high-risk population remains one of the most pressing and unresolved issues in therapy.

Defining “25(OH)D adequacy” shifts higher in CKD patients due to the pathology of SHPT. While general sufficiency is ≥30 ng/mL, studies in non-dialysis CKD suggest this is insufficient for maximal PTH suppression. Observational evidence indicates that PTH levels continue to drop optimally only at 25(OH)D concentrations ranging from 42–50 ng/mL in CKD stages G3–G5. Achieving this higher range (~40–50 ng/mL) is often suggested as a safe and more effective therapeutic target for SHPT in early CKD, as it does not generally increase the incidence of hypercalcemia or hyperphosphatemia. It is acknowledged that while empirical data suggest a higher 25(OH)D range is required for PTH suppression, a definitive, consensus-driven therapeutic target remains a subject of ongoing debate and complexity across the spectrum of CKD stages and patient populations (including transplant recipients) [61].

In advanced CKD (stages G4–G5), the strategy focuses on controlling CKD-Mineral and Bone Disorder (CKD-MBD), which is governed by high FGF23 acting as a barrier: High FGF23 actively suppresses the 1α-alpha-hydroxylase enzyme in the kidney, impairing the conversion of 25(OH)D to active 1,25(OH)2D. This creates a functional active vitamin D deficiency despite adequate precursor levels.

The clinical priority is to correct 25(OH)D deficiency first. However, the critical need to prevent phosphate and FGF23 elevations must be maintained, as high phosphate directly promotes FGF23 synthesis. (The pathological impact of FGF23 is discussed further in Paradigm 3). Furthermore, KDIGO guidelines suggest that calcitriol and its analogues should not be routinely used in non-dialysis CKD(G3a–G5), as calcitriol promotes intestinal absorption of both Ca^2+^ and PO_4_^3−^. Active vitamin D initiation is strictly contingent on safety parameters and is typically held if serum calcium exceeds 9.5 mg/dL or serum phosphorus exceeds 4.6 mg/dL. (The rationale for controlling PO_4_^3−^ and FGF23 is explored in detail in Paradigm 3).

#### 3.2.1. Current Controversies in Correcting Vitamin D Deficiency

Although correcting vitamin D deficiency in CKD is a key guideline recommendation, this seemingly straightforward objective is severely hampered by profound biological and clinical controversies (Reviewed in [47,61]). These challenges stem from a critical disconnect: our standard methods for both assessing vitamin D status and administering supplementation are based on an incomplete and often misleading picture, creating significant hurdles for effective patient management.

#### 3.2.2. The Biomarker Challenge: What Are We Really Measuring?

A core controversy is our inability to accurately assess a patient’s true vitamin D status with standard clinical tools.

(a)Inaccurate Assays and Catabolism: Most clinical labs use assays that cannot distinguish 25(OH)D from its major catabolite, 24,25(OH)_2_D, leading to an overestimation of a patient’s true vitamin D level. A more accurate functional marker—the ratio of 25(OH)D to 24,25(OH)_2_D—can reveal the rate of vitamin D degradation, but measuring this requires liquid chromatography tandem mass spectrometry, which is not widely available. Table 1 compares standard immunoassays and the gold standard LC-MS/MS for measurements of vitamin D metabolites.(b)The “Local Conversion” Blind Spot: Perhaps the most significant limitation is that measuring circulating 25(OH)D completely ignores local, tissue-specific vitamin D metabolism [62]. For instance, tissues like the parathyroid gland can convert cholecalciferol directly to 25(OH)D for their own use [63]. This crucial local activation does not raise systemic 25(OH)D levels and is therefore invisible to our current blood tests, yet it may be essential for local, protective VDR signaling.

#### 3.2.3. The Dosing Paradox and Choice of Agent

Even if vitamin D status could be perfectly assessed, a major paradox exists regarding optimal supplementation. While ergocalciferol supplementation in children with CKD is sufficient to delay the onset of SHPT [64], studies in CKD suggest that high serum 25(OH)D levels (>50 ng/mL) may be needed to suppress PTH [65]. Crucially, large-scale clinical trials in the generally healthy (normal renal function) elderly population (age 70 years or older) have exposed a significant dosing paradox linked to administration frequency. While continuous, lower-dose regimens or monthly bolus doses of cholecalciferol equivalent to 800 IU daily (e.g., 24,000 IU monthly) are associated with beneficial effects on non-vertebral fractures and falls, a subsequent higher monthly bolus of 60,000 IU of cholecalciferol + calcifediol paradoxically increased the incidence of falls compared to the lower dose in the same study. This heightened risk is hypothesized to be a consequence of the intermittent high doses creating unfavorable, high sustained circulating levels of 25(OH)D rather than insufficient therapeutic effect, leading to potential impairments in neuromuscular function. This suggests that the dosing pattern, not just the total dose, dictates safety and efficacy [66].

This adverse effect is likely due to the excessive induction of CYP24A1, the enzyme that rapidly degrades vitamin D, potentially creating a state of functional deficiency despite high intake.

This highlights the critical importance of agent selection, with each having distinct advantages and limitations:(a)Cholecalciferol (D_3_) and Ergocalciferol (D_2_): These are the most common nutritional vitamin D supplements. They are inexpensive and rely on hepatic 25-hydroxylation to raise serum 25(OH)D, with the goal of achieving a normal vitamin D status (>30 ng/mL). When administered in daily doses (typically up to 4000 IU), both forms are considered equally effective [67,68]. However, their pharmacokinetics differ significantly with high-dose, intermittent (bolus) administration. Ergocalciferol (D_2_) has a shorter circulating half-life than cholecalciferol (D_3_) [69]. This is primarily because its metabolite, 25(OH)D_2_, has a lower binding affinity for the Vitamin D Binding Protein (DBP) compared to 25(OH)D_3_. This weaker binding leads to faster metabolic clearance, making bolus D_2_ dosing less efficacious. While overall DBP concentration and genotype do influence the half-life of *all* vitamin D metabolites [70], this fundamental difference in affinity is the key reason for D_2_’s shorter duration in circulation. Furthermore, while high intermittent (bolus) doses are often prescribed to ensure patient compliance, this practice is generally discouraged for two key reasons. First, it carries a risk of potential toxicity, such as transient hypercalcemia. Second, as demonstrated by the work of Armas and co-workers, the hepatic conversion to 25(OH)D is inefficient at high single doses [48]. Their findings indicate that the 25-hydroxylation pathway becomes saturated at intakes that exceed approximately 4000 IU, limiting the effective yield of 25(OH)D from a large bolus.(b)Calcifediol (25(OH)D): Available as standard or extended-release (ER) formulations, calcifediol offers a direct path to raise serum 25(OH)D by bypassing liver activation. Its potency is a key distinction; unlike nutritional vitamin D, calcifediol can directly bind to and activate the VDR [71]. This direct VDR activation, however, increases the risks of hypercalcemia, accelerated catabolism, and the induction of FGF23 (a topic central to Paradigm Shift 3). In stark contrast, clinical trials with the more costly ER-Calcifediol have shown it can effectively raise circulating 25(OH)D and suppress PTH at very high 25(OH)D concentrations [72,73] and also increase serum calcitriol and maintain but not suppress serum PTH [74,75], while avoiding significant elevations in serum FGF23, calcium, or phosphate, presenting it as a potentially safer therapeutic option.(c)The Obesity Factor: A major confounding variable in dosing is obesity. Because vitamin D is fat-soluble, it becomes trapped or sequestered in adipose tissue (Reviewed in [76]). This leads to lower circulating 25(OH)D levels for a given dose, effectively limiting the substrate available for local activation to calcitriol and VDR pleiotropic protective actions in key targets like the cardiovascular system. This is a critical consideration in managing patients with type 2 diabetes and diabetic nephropathy, who are frequently obese and at the highest risk for progressive renal and cardiovascular damage.

#### 3.2.4. A Novel Strategy: 25(OH)D and Calcitriol Synergy

Although there is limited clinical evidence [75], in vitro studies [77], as well as in vivo experimental models of uremia in rats [16], have revealed a previously unrecognized synergy that could offer a safer path to therapy: 25(OH)D synergizes with calcitriol (or its analogs) to enhance VDR activation.

By normalizing circulating 25(OH)D levels, the intracellular concentration of 25(OH)D increases. This synergistic action means that lower doses of active vitamin D or analogs are required to achieve the same therapeutic effect on PTH suppression or other VDR-mediated benefits. This strategy could theoretically overcome vitamin D resistance more safely, offering a crucial advantage for patients in whom hypercalcemia or hyperphosphatemia limits the use of higher doses of active vitamin D.

#### 3.2.5. Corollary: A Shift Toward Functional Dosing

Given these complexities, future strategies for safe and effective vitamin D supplementation may need to shift away from targeting a specific circulating level of 25(OH)D. A more logical approach would be to dose based on a functional biological response.

This would involve supplementation with nutritional vitamin D (or calcifediol) with the goal of achieving a specific clinical benefit—such as PTH suppression, a reduction in proteinuria, or a decrease in inflammatory markers—while meticulously monitoring to prevent the onset of hypercalcemia or hyperphosphatemia. This personalized, response-based approach would prioritize patient safety and biological effect over achieving an arbitrary serum target.

## 4. Paradigm Shift 3: The FGF23-Klotho Axis as the Central Driver of Cardiovascular and Renal Risk

The third, and arguably most critical, paradigm shift has been the discovery of the FGF23-Klotho axis. This has moved the focus from a PTH-centric view to one where the severe imbalance between FGF23 and Klotho is recognized as the dominant driver of cardiovascular and renal progression, especially in diabetic kidney disease (DKD) [78].

### 4.1. Pathophysiological Discoveries and Impact on CKD Progression

#### 4.1.1. The Core Imbalance: FGF23 Resistance and Klotho Deficiency

Physiologically, the primary role of FGF23 is to bind to its essential co-receptor, Klotho, on the surface of renal proximal and distal convoluted tubule cells. This binding is the body’s main signal for promoting urinary phosphate excretion [78,79]. A central feature of CKD is the progressive loss of this renal Klotho [80]. This deficiency, which mimics the severe mineral imbalance of a Klotho knockout [81], renders the kidney resistant to the phosphaturic actions of FGF23, which is a key mechanism for the resulting hyperphosphatemia.

#### 4.1.2. The Vicious Cycle: Drivers of Pathological FGF23 Levels

This renal resistance, combined with hyperphosphatemia, forces a massive compensatory rise in FGF23. The situation is then severely exacerbated by two other factors: (1) the core pathology of CKD, particularly inflammation and iron deficiency, which are potent vitamin D-independent drivers of FGF23 production [82,83], and (2) the use of active vitamin D analogs to treat SHPT, which are powerful stimulators of the FGF23 gene [84]. This severe imbalance—a loss of protective Klotho and this multi-factorial, massive rise in FGF23—leads to systemic toxicity.

#### 4.1.3. FGF23 Toxicity: Klotho-Independent Cardiac and Skeletal Damage

The cardiovascular damage is driven by a recently uncovered Klotho-independent mechanism. In the Klotho-deficient state of CKD, the pathologically high levels of FGF23 can directly bind to and activate FGFR4 receptors on cardiomyocytes [85]. This “off-target” signaling triggers the calcineurin-NFAT pathway, a potent driver of pathological Left Ventricular Hypertrophy (LVH), directly linking the hormone to heart damage. Importantly, while elevated FGF23 is a powerful and independent marker of cardiovascular (CV) risk and mortality in CKD, it does not act in isolation. A balanced view acknowledges that FGF23 contributes to CV morbidity alongside other established and non-traditional risk factors. Indeed, high FGF23 levels frequently coexist with other parameters of CKD-MBD, including hyperphosphatemia (which stimulates FGF23 production), hypo- and hypercalcemia, and SHPT, long-known predictors of CV mortality [86].

FGF23 has also been linked to endothelial dysfunction and arterial stiffness [87,88]. However, traditional CV risk factors, such as hypertension, diabetes, and hyperlipidemia, remain highly prevalent in CKD and cannot be excluded as significant contributors to the overall CV burden. In fact, traditional risk factors are more prevalent in those with higher FGF23 levels.

FGF23’s toxicity also extends directly to the skeleton, contributing to renal osteodystrophy. These same high levels of FGF23 directly impair bone mineralization and healing. This occurs, in part, by FGF23 inducing the Wnt inhibitor Dkk1 in bone cells [89]. Dkk1 then antagonizes the LRP5/6 co-receptor, preventing the activation of Wnt signaling. This inactivation of the Wnt pathway accelerates beta-catenin degradation, thereby preventing the nuclear activation of osteoblast differentiation and bone formation genes.

#### 4.1.4. FGF23 Toxicity: Dismantling the Vitamin D Endocrine System

Making matters worse, the massively elevated FGF23 creates a vicious cycle that actively dismantles the vitamin D endocrine system. FGF23 potently suppresses the synthesis of active vitamin D (by inhibiting CYP27B1 in most tissues, except for the parathyroid glands) [90,91] and simultaneously induces its degradation (by upregulating CYP24A1) [92], which also degrades the 25(OH)D precursor. This ensures that the body loses its own natural protective hormonal system as CKD progresses beyond the kidneys.

#### 4.1.5. The Role of C-Terminal FGF23 Fragments

Furthermore, the problem extends beyond the hormonally active, intact FGF23. The failing kidneys cannot clear FGF23 C-terminal cleavage fragments (cFGF23), which accumulate to extreme levels. These fragments are even stronger predictors of mortality than intact FGF23, reflecting the true burden of the underlying iron deficiency, inflammation, and disease progression [93].

#### 4.1.6. The Protective Role of Klotho: A Systemic Anti-Aging Defense

The counterpoint to this toxicity is the profound protective role of Klotho, which is now understood to be a master regulator of cellular health. As a potent systemic anti-aging hormone [94], soluble Klotho directly combats the drivers of disease progression [95]: it functions as a powerful antioxidant (by activating FoxO and Nrf2 pathways) and an anti-inflammatory agent (by suppressing NF-B and the NLRP3 inflammasome). Furthermore, recombinant soluble Klotho fragments confer protection against renal fibrosis and systemic complications, including uremic cardiomyopathy, primarily by functioning as key antagonists that block pro-fibrotic signaling through binding soluble Wnt ligands [96]. Recent discoveries also show Klotho is a powerful inducer of autophagy—the cell’s essential cleanup process—further defending against cellular senescence. This autophagic pathway mediates the protection of the kidney from the onset of renal lesions and helps delay the transition from acute kidney injury (AKI) to chronic kidney disease (CKD) by supporting renal cell integrity and recovery [97].

In a healthy state, sufficient soluble Klotho can also form a protective FGF23-sKlotho-FGFR1 ternary complex [98], which is thought to antagonize the direct, Klotho-independent toxic effects of FGF23 on the heart. Thus, the FGF23-Klotho axis acts as a ‘dual-edged sword,’ delivering the final blow to an already failing vitamin D endocrine system. Figure 2 provides a comprehensive summary of this systemic collapse, integrating the renal and non-renal defects discussed in Paradigm Shift 2 with the overwhelming toxicity of FGF23.

### 4.2. Therapeutic Implications and Recommendations

#### 4.2.1. The Dilemma (Calcimimetics vs. Vitamin D)

This intricate pathophysiology creates an extraordinary therapeutic dilemma. The challenge from calcimimetics to control SHPT, with lower (if any) increases in FGF23, is valid [99]. However, their efficacy can be variable, potentially because the severe hyperphosphatemia of advanced CKD is known to directly inhibit the CaSR [13], blunting the receptor’s response to these drugs. Furthermore, calcimimetics do not address the profound Klotho deficiency. This deficiency is critical, as it represents the loss of a key systemic anti-aging hormone whose protective, pro-survival actions (as just described) are essential for slowing disease progression.

Significantly, there are important limitations of FGF23-lowering interventional studies. Despite the strong epidemiological association, the question of whether FGF23 is a direct causal mediator or simply a highly sensitive biomarker of disease severity remains under debate. Definite proof that lowering FGF23 improves CV outcomes in humans with CKD is currently lacking in large-scale interventional trials. Indeed, clinical trials focused on reducing phosphate (which secondarily lowers FGF23) often lack the statistical power to show an effect on hard CV outcomes (Reviewed in [100]). Furthermore, complete neutralization of FGF23 in animal models of CKD caused increased mortality, likely by severely impairing the body’s compensatory mechanism for phosphate excretion and leading to profound hyperphosphatemia [101]. This raises safety concerns regarding direct FGF23 blockade in humans with CKD without simultaneously addressing the phosphate load.

It is hypothesized that FGF23 may be adaptive in early CKD (by maintaining phosphate balance) but becomes maladaptive at the extremely high concentrations seen in late CKD. Interventional studies face the challenge of determining the optimal target level for FGF23 in CKD, where the therapeutic goal may be different for early versus late stages.

Conversely, vitamin D therapy, one of the treatments that induces Klotho [102], can worsen the FGF23 burden. Yet, this dilemma is also more complex, as active vitamin D has been shown to decrease cardiac FGFR4 expression [103], potentially limiting the cardiac damage caused by high FGF23 levels. This highlights the deep, sometimes conflicting inter-regulation within this axis.

Importantly, several non-vitamin D agents preserve klotho in CKD: The decline in klotho expression in CKD can be mitigated by therapeutic agents targeting the primary pathogenic drivers: hypertension and inflammation [104,105]. Since Klotho expression is negatively regulated by the RAS, cornerstone therapies as RAS blockers (ACEi or ARB) can prevent or reverse the decline in Klotho expression in the kidney and vasculature by relieving the inhibitory effect that RAS components have on Klotho transcription [106]. Similarly, chronic inflammation and oxidative stress (increased ROS production) are major drivers of Klotho suppression in failing kidneys. Accordingly, compounds as sirtuin activators (e.g., Resveratrol), by activating SIRT1, can upregulate Klotho expression by mitigating oxidative damage and chronic stress [107]. The protective link between metabolic health and Klotho preservation is evidenced by data showing that certain antidiabetic agents (agmantine and pioglitazone) can directly or indirectly induce Klotho expression [108]. Agents such as SGLT2 inhibitors and GLP-1 receptor agonists exert pleiotropic renal and cardiovascular benefits [109]. These agents are thought to indirectly support Klotho preservation by reducing systemic inflammation, improving metabolic control, and attenuating intrarenal RAS activation, all of which are known modulators of Klotho expression.

In addition, mineral homeostasis factors such as phosphate binders can indirectly preserve Klotho by reducing the circulating phosphate load, which in turn decreases the stimulus for FGF23 production. Since FGF23 is the ligand for the Klotho co-receptor, controlling FGF23 is crucial for preventing Klotho exhaustion.

Dietary Intervention modulating specific dietary components (e.g., protein restriction or specific micronutrients) can influence the underlying inflammatory and RAS pathways that regulate Klotho.

#### 4.2.2. The Diagnostic Challenge

Therefore, the future of therapy, particularly in high-risk diabetic CKD, cannot be a simple choice between agents. It demands a combined, nuanced strategy aimed at controlling pathogenic FGF23 (both intact and fragmented) [110] while simultaneously preserving or restoring these essential protective, anti-inflammatory, and pro-autophagic benefits of Klotho. This goal, however, is severely hampered by our current diagnostic limitations, which extend far beyond simply measuring PTH and minerals. We lack affordable, reliable assays to accurately distinguish the whole FGF23 molecule from its fragments. The clinical and prognostic utility of FGF23 is complicated by the different molecular forms of the hormone that circulate in the blood and the limitations of current immunoassays.

FGF23 is naturally a substrate for cleavage by proteases (like Furin), which split the full hormone into two inactive fragments. The full-length, biologically active form of the hormone, intact FGF23 (iFGF23), is the only form that can bind to its co-receptor (Klotho) and receptor (FGFR) to exert its effects (e.g., inhibiting phosphate reabsorption [111].

The measurement of FGF23 is complicated by the presence of both the intact, biologically active hormone (iFGF23) and the C-terminal breakdown products (C-FGF23). Historically, C-FGF23 was considered a largely inert catabolite, with high circulating levels serving merely as a marker of the magnitude of the renal lesion due to the kidney’s impaired clearance function. While high overall FGF23 levels strongly correlate with increased cardiovascular morbidity, direct causality has been historically difficult to establish. Crucially, recent evidence from CKD models has completely overturned this passive view [112]. This work not only demonstrated that iFGF23 is directly pathogenic, contributing to both progression and cardiomyopathy, but also revealed a novel potential therapeutic role for the breakdown product: Blockade of iFGF23 signaling using its natural proteolytic product, C-FGF23, was shown to effectively alleviate kidney and cardiac pathology and improve function in murine models of high endogenous FGF23. This finding repositions C-FGF23 from a passive marker of renal damage to a potential endogenous antagonist, offering a promising, targeted strategy for mitigating FGF23-driven cardiorenal disease.

A key limitation is that C-terminal assays, by measuring both forms, preclude the reliable calculation of molar ratios (e.g., iFGF23/C-FGF23) necessary to fully evaluate the interactions and define the net functional activity of FGF23 signaling.

The inactive molecular C-terminal Fragments (cFGF23) resulting from the cleavage of iFGF23 are non-functional but are cleared by the kidney.

The critical assay challenge lies in the historical use of and reliance on different types of immunoassays: The C-terminal Assays are older assays, which detect the C-terminal fragments (cFGF23) but cannot distinguish between the inactive fragments and the active intact hormone. These assays often show much higher total FGF23 levels, reflecting both active and inactive forms. Instead, Intact Assays specifically detect the active, intact iFGF23.

The accumulation of C-terminal FGF23 fragments was hypothesized to be a superior measure of the overall burden of the disease and the degree of impaired renal clearance. Since the fragments are cleared renally, a high cFGF23 fragment concentration strongly correlates with the severity of the decline in Glomerular Filtration Rate (GFR). Therefore, in late-stage CKD, high cFGF23 may be a more sensitive indicator of global renal dysfunction and the high risk associated with advanced disease than FGF23 alone. This difference necessitates careful consideration when interpreting clinical FGF23 data.

Furthermore, contradictory reports on the accuracy of circulating soluble Klotho have rendered it an unreliable biomarker for tracking disease progression [95]. These challenges, added to the difficulties in evaluating true vitamin D status, mean we are currently attempting to manage a complex, multi-system failure with an incomplete and often-misleading toolkit.

#### 4.2.3. The Call to Action

To move forward, there must be a committed, joint effort by the scientific and medical community. We must develop precise, affordable assays that can finally measure the critical crosstalk within the vitamin D-FGF23-Klotho axis. The ultimate goal is to identify and validate new biomarkers that are most effective at predicting protection from CKD and cardiovascular progression. Only then can we confidently monitor the efficacy of therapies that truly address the lessons from all three paradigm shifts. This vision for a new era of targeted therapy, linking precise biomarkers to combined strategies and optimized outcomes, is conceptualized in Figure 3.

#### 4.2.4. Precision Medicine and Vitamin D in CKD

The “one-size-fits-all” approach to vitamin D is inadequate in CKD, where racial and genetic factors significantly modulate sensitivity and outcomes, advocating for precision medicine.

Regarding racial epidemiology, individuals of African ancestry often exhibit significantly lower circulating 25(OH)D than Caucasians, primarily due to higher melanin content reducing cutaneous synthesis [113]. This suggests the clinical deficiency threshold may need adjustment due to a different lower biological set point in this population.

Regarding genetic/molecular sensitivity, several polymorphisms in key genes modulate vitamin D response and clearance. VDR polymorphisms (e.g., FokI, BsmI) are linked to accelerated CKD progression and differential sensitivity to the PTH-suppressive effects of active vitamin D [114]. Variations in the DBP (vitamin D binding protein) and CYP24A1 (clearance enzyme) genes influence the half-life and availability of vitamin D, contributing to observed racial variability in 25(OH)D levels [115,116]. The interplay of genetics and environment drives clinical risk and disparities: African Americans have a significantly higher incidence of ESRD compared to Caucasians [117]. Differences in vitamin D metabolism and sensitivity may be an unmeasured contributor to these disparities.

The genetic influence on VDR and DBP suggests that fixed-dose supplementation can be suboptimal or excessive. Precision medicine advocates for genotype-guided dosing to maximize the anti-inflammatory and RAS-suppressing benefits of vitamin D and slow CKD progression [115,116].

## Figures and Tables

**Figure 1 ijms-27-00108-f001:**
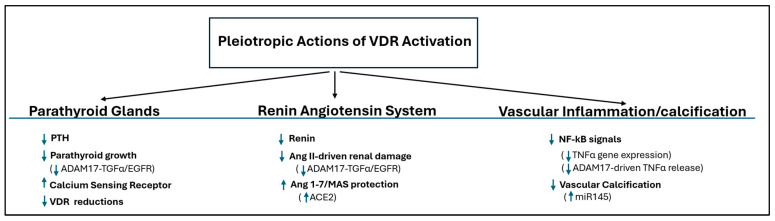
The broad, direct protective effects of vitamin D beyond PTH suppression. Upward arrows indicate increases in the indicated target; Downward arrows indicate reductions in the indicated target.

**Figure 2 ijms-27-00108-f002:**
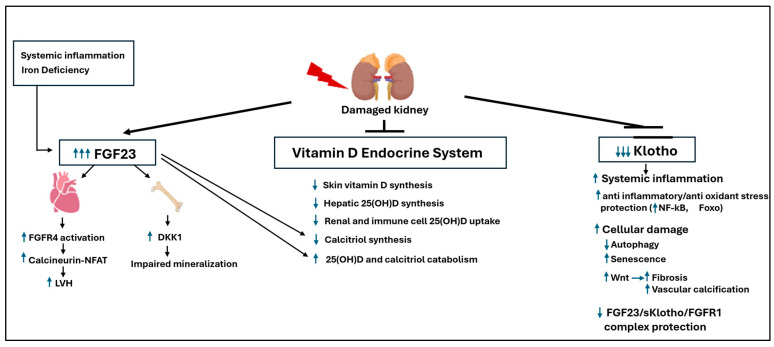
The FGF23-klotho axis: A dual-edged sword in the systemic collapse of the Vitamin D endocrine system: FGF23 is a toxin and klotho is protective, and their balance is key for the prevention of renal and cardiovascular disease progression in CKD. Upward arrows indicate increases in the indicated targets; Downward arrows indicate reductions in the indicated targets.

**Figure 3 ijms-27-00108-f003:**
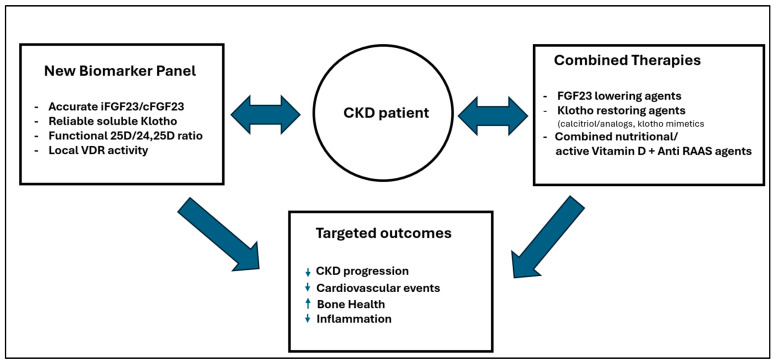
Towards precision medicine: Linking integrated therapies and accurate, affordable biomarkers to guide personalized treatments and achieve true clinical benefits. Upward arrow: increases in the parameters (conditions); downward arrow: decreases in the parameters (conditions).

**Table 1 ijms-27-00108-t001:** Comparison of standard immunoassays and the gold standard LC-MS/MS to measure vitamin D metabolites.

Feature	Standard Immunoassays (e.g., ELISA, Chemiluminescence)	LC-MS/MS (Gold Standard)
Principle	Relies on antibodies binding to 25(OH)D.	Relies on separating and identifying molecules by mass and charge.
Specificity	**Low to Moderate.** Antibodies often cross-react with other vitamin D metabolites (24,25(OH)_2_D and 25(OH)D_2_), leading to overestimation of true 25(OH)D levels.	**High.** Precisely measures individual metabolites separately, providing true concentrations of 25(OH)D_3_ and 25(OH)D_2_.
Matrix Effects	**High.** Susceptible to interference from lipids or other serum components.	**Low.** Pre-separation via LC minimizes matrix interference.
Cost/Throughput	Lower cost, high throughput (suitable for large labs)	Higher initial cost, requires specialized equipment and expertise.
Clinical Standard	Use frequently, but results may lack accuracy for diagnosis	Preferred Standard for accurate diagnosis and clinical trials.

## Data Availability

No new data were created or analyzed in this study. Data sharing is not applicable to this article.

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
