# Peer review of "Current Controversies on Adequate Circulating Vitamin D Levels in CKD"

_ijms, 2025, doi:10.3390/ijms27010108_

Round 1
Reviewer 1 Report
Comments and Suggestions for Authors
The review is timely, conceptually original, and rich in mechanistic insights. The “paradigm shift” structure works well and differentiates this paper from standard CKD-MBD reviews. The discussion of FGF23–Klotho biology and its potential as a therapeutic compass is particularly strong.
Minor suggestions:
- add a dedicated section (e.g. What could ‘adequate’ 25(OH)D mean in CKD?) that summarises: existing thresholds in guidelines; observational evidence by 25(OH)D strata; the trade-off between PTH suppression, extra-skeletal effects and potential harm at higher levels.
-
Regarding the role of FGF-23 in CKD, briefly acknowledge: the contribution of other mechanisms (hyperphosphatemia, arterial stiffness, traditional CV risk factors) and the current limitations of FGF-23-lowering interventional studies in humans.
- The paper nicely frames the tension between FGF23-lowering strategies (e.g. calcimimetics) and VDR activation (which may increase FGF23 but upregulate Klotho). However, some statements (e.g. vitamin D as “the only treatment that induces Klotho”) may be overly absolute given emerging data with other interventions.
Author Response
Response to the Reviewers
We thank the reviewers for their insightful and relevant critiques and suggestions that have greatly improved the contribution of our manuscript.
Response to reviewer 1
We thank Reviewer 1 for their insightful comments and constructive suggestions, which have enhanced the clarity and comprehensiveness of our review. Our point-by-point responses are detailed below.
- Reviewer Comment: Dedicated section. Add a dedicated section (e.g. “What could ‘adequate’ 25OHD mean in CKD”, that summarizes existing guidelines, observational evidence, and the clinical trade-offs between PTH suppression extra-skeletal effects, and potential harm at higher levels.
Our response:
We appreciate the comment regarding the lack of a universal target for 25(OH)D sufficiency in CKD. We maintain that the empirical evidence supports a functional therapeutic range of 42-50ng/mL to achieve maximal PTH suppression in early CKD patients, which is the primary focus of this section.
To further strengthen the discussion and align with the most recent global expert opinion, we have updated this section to explicitly acknowledge the complexity and ongoing debate regarding definitive targets across the entire spectrum of CKD (including pediatric, dialysis, and transplant populations). We now cite the comprehensive European consensus statement on nutritional vitamin D in CKD from Nephrology Dialysis Transplantation (2025), which thoroughly analyzes the knowns and unknowns of D supplementation and confirms the need for individualized, evidence-based strategies that move beyond SHPT management.
- Reviewer Comment: FGF23 Balance: Briefly acknowledge the contribution of other mechanisms (hyperphosphatemia, arterial stiffness, traditional CV risk factors) and the current limitations of FGF23-lowering interventional studies in humans.
Our response:
We agree with the reviewer that a balanced perspective on FGF23 is essential. We have expanded our discussion within Paradigm 3 to explicitly acknowledge that the FGF23-associated cardiovascular risk is complicated by the contribution of coexisting factors (hyperphosphatemia, arterial stiffness, and traditional risk factors) and have included a necessary section detailing the current scientific limitations of FGF23-lowering interventional studies in humans.
- Reviewer Comment: Klotho Induction: Some statements (e.g., vitamin D as "the only treatment that induces Klotho") may be overly absolute given emerging data with other interventions.
Our response:
We thank the reviewer for this important comment and agree that the statement defining active vitamin D as the 'only' inducer of Klotho is an oversimplification. We have modified this statement in the revised manuscript to reflect that vitamin D is a key regulator, but that other therapeutic interventions used in management also strongly influence Klotho expression. Specifically, we now acknowledge that RAS blockers (ACEi/ARB) and several anti-inflammatory agents have been shown to prevent the decline or induce the expression of Klotho in preclinical and clinical settings, mainly by mitigating the effects of hypertension and oxidative stress, which are major determinants of Klotho reduction.
Reviewer 2 Report
Comments and Suggestions for Authors
This review offers a timely and comprehensive synthesis of the evolving understanding of vitamin D's role in chronic kidney disease (CKD), structured around three major paradigm shifts. It effectively integrates pathophysiological mechanisms, clinical evidence, and ongoing therapeutic controversies, with particular emphasis on the interplay among vitamin D, secondary hyperparathyroidism (SHPT), and the FGF23-Klotho axis. The manuscript provides added value by identifying key diagnostic and therapeutic challenges, including the limitations of current biomarkers and the potential necessity of integrated therapeutic approaches—insights that may inform future research directions. The presentation is clear, logically organized, and supported by relevant figures; however, certain sections would benefit from minor clarifications to enhance readability for a broader audience. Overall, the review is of high scholarly quality but requires minor revisions to improve factual accuracy, update references, and address editorial refinements.
- Section 2.1.1(d): The description of parathyroid desensitization through suppression of the VDR by the C/EBPβ/LIP ratio is accurate; However, incorporating updated evidence to strengthen the supporting documents for this claim.
- Section 4.1.5: The role of cFGF23 fragments as more robust predictors of mortality is noteworthy; it is recommended to include a brief discussion of assay-related challenges—such as the difficulty in distinguishing between intact FGF23 and its fragments—to better align with the diagnostic limitations outlined in Section 4.2.2.
- Section 3.2.2: The discussion of the "biomarker challenge"—for example, the overestimation of 25(OH)D levels due to cross-reactivity with 24,25(OH)₂D in immunoassays—is well articulated; however, the inclusion of a simple comparative table outlining standard immunoassays versus LC-MS/MS methods would enhance clarity for non-specialist readers.
Author Response
Response to Reviewer 2
We thank Reviewer 2 for the insightful and constructive comments, which have allowed us to clarify complex technical points and strengthen the molecular and diagnostic aspects of the manuscript. Our point-by-point responses are detailed below.
- Section 2.1.1(d): C/EBPβ/LIP Ratio and VDR Suppression
Reviewer Comment: The description of parathyroid desensitization through suppression of the VDR by the reduced C/EBPβ/LIP ratio is accurate; however, incorporating updated evidence to strengthen the supporting documents for this claim.
Our Response:
We have significantly updated the discussion on the C/EBPbeta mechanism. We now explicitly integrate the molecular evidence demonstrating how inflammatory stimuli (e.g., via the LPS pathway) drive the translational switch to favor the inhibitory LIP isoform. We expand the significance of this finding by discussing how LIP accumulation is linked to Endoplasmic Reticulum (ER) stress-driven apoptosis—a critical pathway for non-skeletal damage (e.g., vascular calcification propensity) in CKD. This update addresses the upstream driver and the resultant systemic pathology, substantially strengthening the section as requested.
We have further strengthened this section by noting that vitamin D deficiency itself increases ER stress—the same mechanism amplified by LIP—thereby establishing that the LIP/ER stress pathway is directly triggered by the CKD/D Deficiency state, providing a powerful mechanistic explanation for the observed systemic damage.
- Section 4.1.5: Fragments and Assay-Related Challenges
Reviewer Comment: The role of cFGF23 fragments as more robust predictors of mortality is noteworthy; it is recommended to include a brief discussion of assay-related challenges—such as the difficulty in distinguishing between intact FGF23 and its fragments—to better align with the diagnostic limitations outlined in Section 4.2.2.
Our Response:
We appreciate the comment regarding the strong correlation between high FGF23 and cardiovascular morbidity, where direct causality has often been debated. To fully address this, we have significantly strengthened our discussion on the clinical relevance of distinguishing between iFGF23 and C-FGF23 by incorporating the critical NEW & NOTEWORTHY finding. This seminal work provides the necessary mechanistic evidence:
- It establishes iFGF23 as directly pathogenic for both kidney and heart in CKD models, thus confirming causality.
- More importantly, it completely overturns the traditional view of C-FGF23 as a mere inert catabolite or passive marker of renal lesion magnitude. The study demonstrates that C-FGF23 acts as a natural antagonist to iFGF23 signaling, successfully reversing cardiorenal pathology when administered.
This translational finding provides essential justification for our chapter's focus on the active hormone and its therapeutic blockade. We have fully incorporated this contrast into the discussion to highlight the paradigm shift—from viewing C-FGF23 as a passive marker of damage to a potential endogenous therapeutic agent.
- Section 3.2.2: Biomarker Challenge and Table
Reviewer Comment: The discussion of the "biomarker challenge"—for example, the overestimation of 25(OH)D levels due to cross-reactivity with 24,25(OH)D in immunoassays—is well articulated; however, the inclusion of a simple comparative table outlining standard immunoassays versus LC-MS/MS methods would enhance clarity for non-specialist readers.
Our Response:
We agree with the reviewer that a clear visual aid would enhance the discussion of the "biomarker challenge" for non-specialist readers. We have added a simple comparative table to Section 3.2.2 outlining the key differences between standard immunoassays and the gold-standard LC-MS/MS method. This table visually emphasizes that the cross-reactivity of immunoassays with metabolites like 24,25(OH)2D leads to the 25(OH)D overestimation that we discuss, thereby enhancing the clarity of our statement. The inclusion of this table should greatly assist in differentiating the analytical limitations of the assays discussed.
Reviewer 3 Report
Comments and Suggestions for Authors
This is a well written review of the clinical paradigm shifts and controversies about circulating Vitamin D and CKD. The manuscript is well organized and brings a comprehensive, critical analysis of evidence in the past three decades on the three paradigm shifts. However, I would like to raise two questions that might help make this review more readable.
- For the first paradigm shift, are there literature in the recent years that support and update the understanding of pathophysiology and outcomes related to metabolism of Vitamin D in CK patients and CKD progression? Most of the citations for this section are older than 15 years. Have any review articles analyzed this paradigm shift in a similar lens?
- In the context of precision medicine, are there studies about the racial differences and genetics in the sensitivation of Vitamin D and any impact of such difference to CKD progression at molecular and clinial levels?
Author Response
Response to Reviewer 3
We thank Reviewer 3 for their detailed and insightful comments. We agree that grounding our review in the most contemporary literature and incorporating the complexity of personalized medicine are crucial for the manuscript's scientific impact. Our point-by-point responses are detailed below.
- Section 1 (Foundational Paradigm): Updating Outdated Citations
Reviewer Comment: For the first paradigm shift, are there literature in the recent years that support and update the understanding of pathophysiology and outcomes related to metabolism of Vitamin D in CKD patients and CKD progression? Most of the citations for this section are older than 15 years. Have any review articles analyzed this paradigm shift in a similar lens?
Our Response:
We thank the reviewer for highlighting the necessity of updating the citations in our foundational "first paradigm shift" section. We recognize that while the core metabolic principles remain valid, the field has been significantly refined by recent large-scale observational studies and comprehensive reviews. We have conducted a thorough literature search (focusing on sources published since 2018) and replaced the outdated citations with contemporary review articles and meta-analyses. This ensures that our established principles regarding:
- Altered Metabolism: The FGF23-driven impairment of renal synthesis (suppression of 1-alpha-hydroxylase expression) and induction of catabolism are now supported by this comprehensive current review on CKD-MBD.
- High Prevalence & Outcomes: The link between vitamin D deficiency and adverse outcomes (mortality, CV events, CKD progression) and the controversies among clinical trials is documented by a critical evaluation of a fully updated meta-analyses.
- Pleiotropic Actions: The non-skeletal effects, including VDR suppression of the RAS and anti-inflammatory actions, are supported by current literature on vitamin D renoprotective role in additional recent reviews by experts in the respective areas.
- Section 2 (New Section): Racial Differences and Precision Medicine
Reviewer Comment: In the context of precision medicine, are there studies about the racial differences and genetics in the sensitivation of Vitamin D and any impact of such difference to CKD progression at molecular and clinical levels?
Our Response:
We agree that integrating the concepts of precision medicine, genetics, and race is critical for a contemporary review. We have added a dedicated section to address the impact of racial differences and genetics on vitamin D sensitivity and CKD progression.
- Racial Differences: We discuss that individuals of African ancestry often exhibit lower circulating 25(OH)D levels due to melanin's effect on cutaneous synthesis, which may necessitate a different interpretation of the clinical deficiency threshold.
- Molecular Level: We discuss the role of genetic polymorphisms in key regulatory genes, including the Vitamin D Receptor (VDR) (e.g., FokI, BsmI) and metabolic/transport genes (CYP24A1/DBP). These variations modulate the cellular response and metabolic clearance of vitamin D.
- Clinical Level: We explicitly link these differences to the disparities in CKD outcomes and advocate for moving toward a genotype-guided dosing strategy to maximize the anti-inflammatory and RAS-suppressing benefits of vitamin D in specific populations.
Round 2
Reviewer 3 Report
Comments and Suggestions for Authors
Appreciate the opportunity of reviewing the revision and author's response to my first review. This draft is well strengthen with the revision that has addressed my comments in the initital review. I do not have further questions or suggestions. Thus, I highly recommend this paper be accepted for publication.